# Biomechanical Posture Analysis in Healthy Adults with Machine Learning: Applicability and Reliability

**DOI:** 10.3390/s24092929

**Published:** 2024-05-04

**Authors:** Federico Roggio, Sarah Di Grande, Salvatore Cavalieri, Deborah Falla, Giuseppe Musumeci

**Affiliations:** 1Department of Biomedical and Biotechnological Sciences, Section of Anatomy, Histology and Movement Science, School of Medicine, University of Catania, Via S. Sofia n°97, 95123 Catania, Italy; federico.roggio@unict.it; 2Department of Electrical Electronic and Computer Engineering, University of Catania, Viale A. Doria 6, 95125 Catania, Italy; sarah.digrande@phd.unict.it (S.D.G.); salvatore.cavalieri@unict.it (S.C.); 3Centre of Precision Rehabilitation for Spinal Pain (CPR Spine), School of Sport, Exercise and Rehabilitation Sciences, University of Birmingham, Birmingham B15 2TT, UK; d.falla@bham.ac.uk; 4Research Center on Motor Activities (CRAM), University of Catania, Via S. Sofia n°97, 95123 Catania, Italy; 5Department of Biology, Sbarro Institute for Cancer Research and Molecular Medicine, College of Science and Technology, Temple University, Philadelphia, PA 19122, USA

**Keywords:** biomechanics, machine learning, posture analysis, reliability, principal component analysis, cluster analysis, musculoskeletal disorders, ergonomics, kinesiology

## Abstract

Posture analysis is important in musculoskeletal disorder prevention but relies on subjective assessment. This study investigates the applicability and reliability of a machine learning (ML) pose estimation model for the human posture assessment, while also exploring the underlying structure of the data through principal component and cluster analyses. A cohort of 200 healthy individuals with a mean age of 24.4 ± 4.2 years was photographed from the frontal, dorsal, and lateral views. We used Student’s *t*-test and Cohen’s effect size (d) to identify gender-specific postural differences and used the Intraclass Correlation Coefficient (ICC) to assess the reliability of this method. Our findings demonstrate distinct sex differences in shoulder adduction angle (men: 16.1° ± 1.9°, women: 14.1° ± 1.5°, d = 1.14) and hip adduction angle (men: 9.9° ± 2.2°, women: 6.7° ± 1.5°, d = 1.67), with no significant differences in horizontal inclinations. ICC analysis, with the highest value of 0.95, confirms the reliability of the approach. Principal component and clustering analyses revealed potential new patterns in postural analysis such as significant differences in shoulder–hip distance, highlighting the potential of unsupervised ML for objective posture analysis, offering a promising non-invasive method for rapid, reliable screening in physical therapy, ergonomics, and sports.

## 1. Introduction

The structure and function of the body provide the potential for achieving and maintaining an appropriate posture; however, the persistence of poor postural habits can lead to discomfort, pain, or disability [1]. Many methods are available to evaluate posture, each of them with strengths and weaknesses [2]. Optoelectronic motion capture systems are considered the gold standard for human movement analysis [3], although they are expensive and commonly limited to laboratory environments. On the other hand, smartphone applications are low-cost and portable [4,5], but their reliability may be lower [2].

Artificial Intelligence (AI), particularly machine learning (ML), has revolutionized healthcare data analysis [6], supporting the diagnosis of various conditions like gait disorders [7], Parkinson’s disease [8], stroke [9], and osteoarthritis [10]. Traditional human motion analysis methods are rapidly evolving due to advancements in ML and computer vision. For example, previous studies have explored ML models for identifying postural abnormalities in Parkinson’s disease patients using Microsoft Kinect data [11]. The development of sophisticated pose estimation models has revolutionized anatomical landmark extraction from images and videos, enabling a new era of ML-driven movement analysis. However, applying ML human pose estimation models specifically for posture assessment remains a relatively unexplored area [12]. Traditional posture assessment methods often suffer from subjectivity and low reliability due to visual inspection [13]. This inconsistency underscores the need for an objective, AI-driven method that minimizes clinician bias, which is crucial given the role of posture in musculoskeletal disorder prevention [14]. This presents a significant opportunity for innovative research in this field [12]. Recently, many digital alternatives have emerged as accessible methods for human pose estimation, such as MediaPipe [15], OpenPose [16], MoveNet [17], and PoseNet [18]. Their strength lies in the possibility of performing objective and reproducible analyses with a simple video or photo achieved in any environment [19,20]. Such methods are being employed in different scenarios. For example, while traditional ergonomic risk assessment often relies on human observation, recent ML-based methods have demonstrated the value of analyzing a worker’s musculoskeletal load based on relative body part positions, highlighting their applicability in work-related scenarios [21]. Additionally, previous research has compared the effectiveness of OpenPose for gait analysis directly with marker-based motion capture systems. This study concluded that the ML approach outperformed traditional methods, demonstrating its potential as a cost-efficient alternative for video analysis in out-of-laboratory environments [22]. Furthermore, studies using OpenPose have shown promising results in applying a deep learning pipeline to differentiate neurological gait patterns between Parkinson’s disease and multiple sclerosis [23]. The development of in-home gait monitoring tools could be a valuable resource for both diagnosis and clinician support in telemedicine systems.

MediaPipe Pose is a powerful ML algorithm from the Google research team, designed to accurately track human body poses. It estimates 33 landmarks across the body from 2D photos or videos, enabling detailed motion tracking by determining their 3D positions [24,25]. To ensure the accuracy of its 3D pose estimations, MediaPipe uses ground truth data obtained through a statistical 3D human body model (GHUM), built using a large dataset of human shapes and motions [26]. This model is then fitted to 2D pose data and further refined with real-world 3D keypoint coordinates. During this process, the shape and pose variables of the GHUM are optimized to align with image evidence for maximum precision. The validity of the MediaPipe joint inference tracking technique has been thoroughly investigated. Lafayette et al. [27] conducted a quantitative evaluation of its angular estimation capabilities by comparing it to a gold-standard Qualisys motion capture system [28]. They found an excellent absolute relative clinical error and a strong correlation with Qualisys, with mean Pearson’s coefficients of 0.80 and 0.91 for lower and upper limb movements. Further validation studies on MediaPipe performance come from its ability to categorize exercise postures with 100% precision [29] and detect human movements in non-standard videos with over 90% accuracy [30].

Principal component analysis (PCA) is a powerful dimensionality reduction technique widely used in machine learning and movement analysis [31,32]. It offers a non-reductionist approach to biomechanics, minimizing investigator bias and enriching the understanding of body part interactions [32,33]. Clustering analysis is another valuable method for discovering new patterns in multidimensional data by organizing them into clusters based on similarities [34]. It has applications in biomechanics, including gait analysis [35], movement kinematics [36], and injury prevention [37]. The rationale for using ML algorithms like PCA and cluster analysis in posture assessment lies in their ability to analyze high-dimensional data, revealing patterns and relationships that might be overlooked by traditional methods.

The significant potential of these ML approaches makes their transition into clinical settings crucial, especially considering the limited number of studies that have attempted this to date [19,38]. Therefore, the aim of the current study is two-fold. First, it aims to demonstrate the applicability and reliability of an ML approach for analyzing posture, with the goal of providing normative data on healthy men and women. Subsequently, it aims to offer new insights into posture by applying PCA and clustering methods to the data. This dual approach aims to enhance the depth of posture analysis in the clinical context.

## 2. Materials and Methods

For this cross-sectional study, a total of 250 volunteers were initially recruited at the Research Center on Motor Activities (CRAM) of the University of Catania, Italy, from March 2023 to June 2023. After applying exclusion criteria for musculoskeletal and neurological pathologies, or musculoskeletal trauma within the past six months, 200 healthy participants, consisting of 84 men and 116 women, were included, aged between 18 and 30 years (mean age 24.4 ± 4.2 years). Participants were requested to wear minimal clothing, such as shorts and a t-shirt for men, or a sports bra and shorts for women. The required sample size was calculated using G*Power. To ensure adequate statistical power and account for potential data loss, we aimed for a slightly larger sample size from 176 to 200 participants.

Photos were collected while the participants adopted the anatomical zero position [39]; this involved standing upright, facing forward, with arms resting at their sides, hands facing the body, and forearms midway between supination and pronation. The feet were aligned, spaced 10 cm apart, and positioned at a 20° total out-toeing angle. They were asked to stand still 1.5 m distant from a smartphone with a 50 MP 1/1.56″ sensor camera mounted on a tripod while collecting the photos. Each measurement was collected in the morning, from 9 a.m. to 12 a.m. To ensure the reliability of the study, a subgroup of 90 participants underwent the same procedure after one week. This study received approval from the Scientific Committee of the University of Catania’s Research Center on Motor Activities (Protocol n.: CRAM-035-2023, 15 March 2023), and was conducted in accordance with the Declaration of Helsinki. Participants signed written informed consent forms to agree to participate in the study.

### 2.1. Pose Estimation

A total of 33 anatomical landmarks were predicted using MediaPipe Pose with a BlazePose and MobileNetv2 Convolutional Neural Network architecture [40]. This model detects human bodies, analyzes images, produces heatmaps and offsets, and identifies body landmarks, as shown in Figure 1. Body orientation is determined automatically by identifying the hip midpoint and the angle of the shoulder–hip line. To obtain 3D projections of each anatomical landmark, we extracted their x, y, and z coordinates; then, we calculated joint angles between body segments using the circular mean of data from frontal and dorsal photos to improve accuracy. Further, we determined vertical and horizontal inclinations of vectors between landmarks. Body vector lengths were calculated as distances between landmarks, e.g., shoulder–elbow, using pixel distance (pd) as measurement unit. The lateral photo was used for neck and trunk inclinations. For detailed calculation methods, refer to Table A1.

### 2.2. Data Analysis

The data underwent three layers of analysis. We used Python for building our algorithm model and R Project for Statistical Computing (Vienna, Austria) for the statistical analyses. The Shapiro–Wilk test verified the normality of the data, while Student’s *t*-test and Mann–Whitney U test identified any significant differences between men and women, considering results with a *p*-value < 0.05 to be significant. Cohen’s d measured the effect size between the two groups. Pearson’s correlation coefficient (r) assessed the association between height and postural variables, considering relevant only the results with *p*-value < 0.05 and r > 0.45. Then, we used the Intraclass Correlation Coefficient (ICC) (3,k) to assess the reliability of this method, along with the Standard Error of Measurement (SEM) and Minimal Detectable Change (MDC).

### 2.3. Principal Component and Clustering Analyses Methods

We conducted multivariate statistical techniques to examine whether the results could highlight other specific patterns within the sample. Given the presence of 23 variables (see Table A1 for full details), PCA was utilized to reduce dimensionality by transforming correlated variables into uncorrelated principal components, maximizing the explained variance with each component. After conducting PCA, we applied the clustering algorithms, k-means, mean-shift, Ward’s method, complete linkage, and average linkage, to identify natural groupings within the data. The rationale for employing a variety of methods was to ensure a comprehensive exploration of the dataset to determine the strengths of each technique and observe different patterns and relationships within the data. The silhouette score was used to assess the quality of the clustering algorithms.

## 3. Results

The acquired postural parameters, representing the circular mean of 3D landmarks from frontal and dorsal photos (Table 1, Figure 2), showed significant differences between men and women. Significant anthropometric differences were observed between men and women for age (*p* < 0.05), height (*p* < 0.001), and weight (*p* < 0.001). Men had a mean age of 25.9 ± 5.2 years, a mean body weight of 68.2 ± 10.8 kg, and a mean height of 175 ± 6.4 cm. Women had a mean age of 24.1 ± 4.8 years, a mean body weight of 54.4 ± 3.9 kg, and a mean height of 163 ± 6.1 cm. Notable differences were found in shoulder adduction, elbow extension, hip adduction/extension, and ankle flexion, but not knee flexion or varus/valgus angles. The greatest effect size was observed for hip adduction d = 1.67, and the smallest for knee extension d = 0.01. The greatest effect size for vertical inclinations was in trunk forward inclination (d = 0.66), with no significant difference in leg inclination. No significant differences or large effect sizes were observed in horizontal inclinations. Finally, all vectors showed statistical significance with a large effect size. The shoulder–hip difference demonstrated a valuable effect size of d = 1.44. Concerning the correlation analysis, we observed a significant association between height and posture specifically for the following variables: torso vector (r = 0.60, *p* < 0.001), total arm vector (r = 0.496, *p* < 0.001), total leg vector (r = 0.530, *p* < 0.001), and shoulder–hip difference (r = 0.473, *p* < 0.001).

### 3.1. Test–Retest Reliability

The ICC (3,k) was used to assess the reliability of measurements when repeated on the same sample after a week. Table 1 shows excellent reliability across all measurements, with ICC (3,k) values ranging from 0.67 to 0.95.

### 3.2. Principal Component Analysis and Clustering Methods

We fed the data into the PCA algorithm and identified four components. To ensure the representation of 90% of the data variance, we focused on the first two main components, which had variance ratio values of 0.815 and 0.093. Both the elbow method and the silhouette analysis suggested the presence of two cluster groups (CG1, CG2). The majority of clustering algorithms produced similar silhouette scores (approximately 0.55) and cluster divisions. However, the mean-shift method yielded the highest score (0.568), and we therefore adopted its cluster divisions for further analyses, as shown in Figure 3. Inferential statistics were used to analyze the data divided by the mean-shift method (Table 2). The cluster analysis highlighted a significant difference in body vector lengths, particularly in the shoulder–hip difference: CG1 = 57.5 ± 7.0 pd while CG2 = 86.6 ± 10.7 pd, with a large effect size of d = 3.21.

## 4. Discussion

Postural assessment plays a fundamental role in clinical [41], sport [42], and ergonomic evaluation [43]. This study investigated using ML to enhance postural analysis, established normative posture data for healthy men and women, and explored interesting posture patterns using PCA and cluster analysis. Our findings offer insights into the reliability of ML-driven posture analysis and its potential to reveal previously unidentified relationships within postural data.

We expected a significant difference in joint angles and no significant difference in horizontal and vertical inclinations. While postural asymmetries are prevalent in various conditions such as low back pain [44], stroke [45], cerebral palsy [46], Parkinson’s disease [47], and scoliosis [48], we recognized that our sample, comprised of healthy participants, would not show substantial asymmetry. These observations differ from our previous work on posture assessment using rasterstereography, where we assessed only the trunk [49]. This difference suggests that the ML approach may be more sensitive in detecting subtle imbalances in healthy individuals. The body angles varied by sex, with a pronounced effect size observed for the shoulder adduction, elbow extension, hip adduction/extension and ankle flexion. In contrast, the knee varus/valgus, hip extension, and neck and trunk inclinations, while statistically significant, exhibited a medium effect size. Men demonstrated higher values of shoulder adduction and elbow extension. However, since these angles influence each other, it is challenging to identify the exact reason for this sex difference. For the neck, a more pronounced inclination was observed in women. This difference, with a medium effect size (d = −0.55), could be related to the varied behavior of the neck during daily activities. For example, Tierney et al. [50], found that women have lower neck isometric strength, neck girth, and head mass compared to men when responding to an external force. This subtle difference might influence neck posture, possibly resulting in a generally more forward head position for women.

While some studies have validated already the accuracy of ML algorithms in identifying joint center locations with a precision of 30 mm or less [51,52], the use of ML in assessing human posture is a growing field with limited studies to date. Moreira et al. [53] recently published an article employing a PoseNet API-based method to assess human posture in a sample of 20 adults. Unlike our study, they only calculated the tilt, i.e., horizontal inclination of five anatomical landmarks. They observed a horizontal head tilt of 3.77° ± 2.92, shoulder tilt of 2.18° ± 1.56, hip tilt of 1.58° ± 0.86, ankle tilt of 1.67° ± 0.98, right knee tilt of 163.93° ± 4.03, and left knee tilt of 165.94° ± 3.56. Comparing these results with ours, we noted discrepancies for the head, shoulders, and knee tilts. Notably, their definition of “knee tilt” corresponds to the valgus/varus position of the knee. However, their reported values of approximately 14–16° (obtained by subtracting their results from the flat angle of 180°) seem questionable, as other studies have indicated that a knee angle of −3° or +3° is indicative of a valgus or varus knee, respectively [54,55]. Despite their higher values, the validity of their results is uncertain due to the heterogeneous nature of their sample, ranging from 12 to 66 years old. Our findings for the knee varus/valgus angle align with the neutral range. The lack of similar studies using ML methods for posture analysis makes direct comparisons with existing research challenging. While we found some similarities with two studies using a mobile application, detailed comparisons are difficult [4,56]. These apps rely on specific positions of the body landmarks, whereas human pose estimation algorithms are based on 3D joint center locations [57].

In order to assess the reliability of this method, we tested a sample of participants twice and evaluated the ICC (3,k). We found substantial-to-excellent agreement for both horizontal and vertical inclinations, while measures of the body joints demonstrated excellent agreement. The reliability of MediaPipe has previously been explored in other studies. For example, Ota et al. [58] assessed the reliability using a motion capture system during bilateral squat movements and found ICC scores ranging between 0.92 and 0.96. Saiki et al. [38] conducted two studies on the reliability of their method in patients with knee osteoarthritis. They initially demonstrated excellent test–retest reliability (ICC (1, 1)  =  1.000) and substantial agreement with radiography (ICC (2, 1)  =  0.915) for measuring the hip–knee–ankle angle. A subsequent study on knee range of motion after total knee arthroplasty further showed high ICC values for knee extension and flexion (ICC = 1.000 for both). Their findings were consistent with both X-ray and goniometry measurements (ICC range: 0.963–0.994) [59]. Also, Latreche et al. [60] confirmed the valid results of MediaPipe for specific rehabilitation movements. They reported ICC values of 0.96, 0.99, 0.99, and 0.99 for shoulder abduction, adduction, extension, and flexion movements, respectively. Furthermore, Hii et al. [61] compared the gait analysis accomplished with MediaPipe and Vicon cameras, finding good-to-excellent ICC agreement in all temporal gait parameters except for double support time, which exhibited an ICC > 0.50.

In this study, we analyzed the length of body vectors. While limited in reflecting real-world measurements, they offer insight into underlying anatomical variations. The magnitude of the correlation coefficients (ranging from 0.473 to 0.60) indicates a moderately strong relationship between height and posture variables. This positive correlation signifies that as height increases, there tends to be a corresponding change in the length of body vectors. As expected, we observed a less pronounced than anticipated shoulder–hip difference between sexes (men: 83.8 ± 14.9 pd; women: 63.4 ± 13.3 pd), aligning with well-established sex differences in body lengths [62]. When we performed cluster analyses, we expected it to primarily differentiate participants based on their sex. However, we found a marked distinction in body vector lengths between the two groups formed by the analysis (CG1 and CG2), and this distinction was independent of the sex of participants. Both groups contained a mix of men and women, demonstrating the ability of the cluster analysis to identify a pattern of posture variation beyond the traditional sex-based anatomical differences. This distinction was much more pronounced than the subtle differences observed for body angles and inclinations within the cluster analysis. Notably, the arm and torso lengths differed between groups: 233.4 ± 16.2 pd (CG1) vs. 295.9 ± 21.6 pd (CG2), with an effect size of 3.28; and 241.2 ± 14.1 pd (CG1) vs. 307.7 ± 23.4 pd (CG2), with an effect size of 3.44. The most prominent difference influencing PCA was the shoulder–hip differential: 57.5 ± 7.0 pd (CG1) vs. 86.6 ± 10.7 pd (CG2), with an effect size of 3.21. This difference highlights two distinct postural types similar to the somatotype classification. Olds et al. [63] utilized 3D anthropometry with 301 adults to identify body shape clusters, aligning their findings with traditional somatotype classifications [64], i.e., endomorphic: larger widths, shorter limbs; ectomorphic: longer limbs, narrower hips. Our study followed a different approach, employing an unsupervised ML method to discern two distinct postural categories. However, these categories appear to echo the somatotype trends: CG1 includes individuals with shorter body segments and similar shoulder and hip widths; CG2 includes individuals with longer body segments with notably wider shoulders compared to hips.

This data-driven approach offers a novel perspective on postural assessment. Our findings highlight the potential of ML models to provide fast, reliable posture analysis. In clinical scenarios, this could enhance the understanding of anatomical variations, support personalized interventions in physical therapy and ergonomics, potentially aid preventive screening, and help define the boundaries between correctable and pathological posture. Furthermore, it could help coaches evaluate the effectiveness of athletic movements [65], optimize training, and prevent posture-related injuries [66,67]. Requiring minimal clothing removal, this ML-based method reduces subjectivity in traditional postural analysis and could guide physiotherapy, remote rehabilitation, and pathology management [68,69]. Additionally, the use of PCA and clustering suggests that posture classification may be possible regardless of sex. While cluster analysis provides valuable insights into sport tactics [70], its application to athletes’ posture data may represent an innovative approach. This could help identify physiological characteristics closely linked to specific sports, revealing how aspects of the human body directly impact athletic performance.

This research presents certain limitations. First, our analysis focused on a homogenous sample. While we found significant differences, a more diverse population with a wider age range (e.g., 40–70) could yield more robust findings and highlight age-specific conditions, as well as highlighting any age-specific postural differences. Second, requiring a specific posture for the photos might potentially obscure certain postural alterations. Future research should encompass a broader demographic to highlight more generalizable postural conditions, as well as the analysis of dynamic postures. Furthermore, this approach should be tested in the context of specific musculoskeletal disorders, e.g., lower back pain, and other orthopedic or neurological conditions.

## 5. Conclusions

This study introduces a novel ML approach with significant clinical potential for postural analysis, offering distinct classification and reduced subjectivity. This efficient, non-invasive method could enhance personalized treatment in physical therapy and ergonomics. Our research confirms its applicability and reliability in assessing healthy adult posture, providing normative data. We identified highly reliable postural parameters and highlighted sex-related differences. Importantly, cluster analysis revealed postural characteristics independent of sex, such as limb length and shoulder–hip width differences. These findings suggest potential new opportunities for ML-driven posture classification.

## Figures and Tables

**Figure 1 sensors-24-02929-f001:**
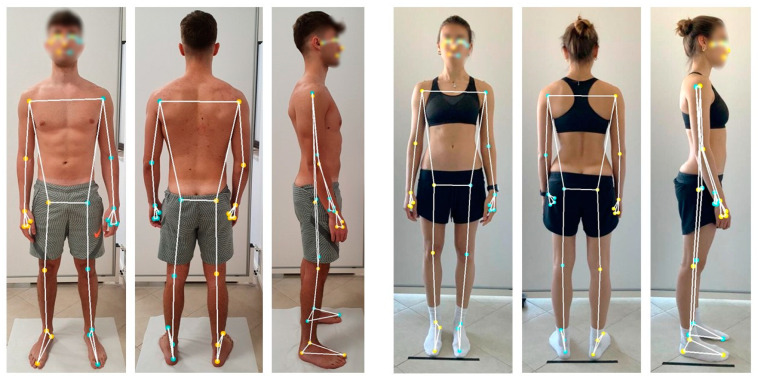
Pose estimation of the collected photos, visualization of the superimposed skeletal model highlighting the anatomical landmarks on frontal, dorsal, and lateral photo views of a woman and a man.

**Figure 2 sensors-24-02929-f002:**
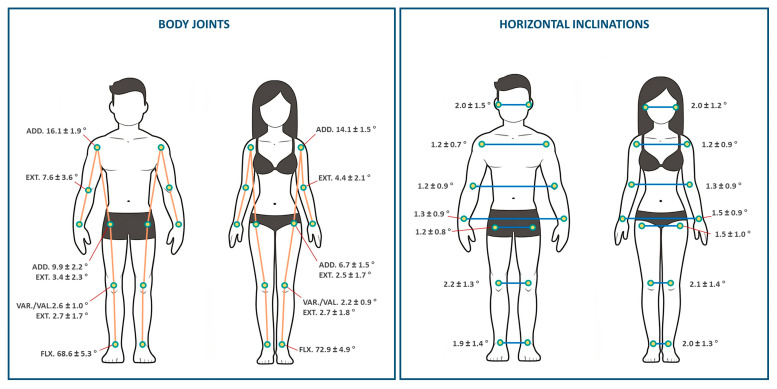
Angle results of the machine learning posture analysis with detailed comparison of body joint angles and horizontal inclinations for men and women, as indicated by mean values ± standard deviations. (ADD: adduction, EXT: extension, VAR/VAL: varus/valgus, FLX: flexion).

**Figure 3 sensors-24-02929-f003:**
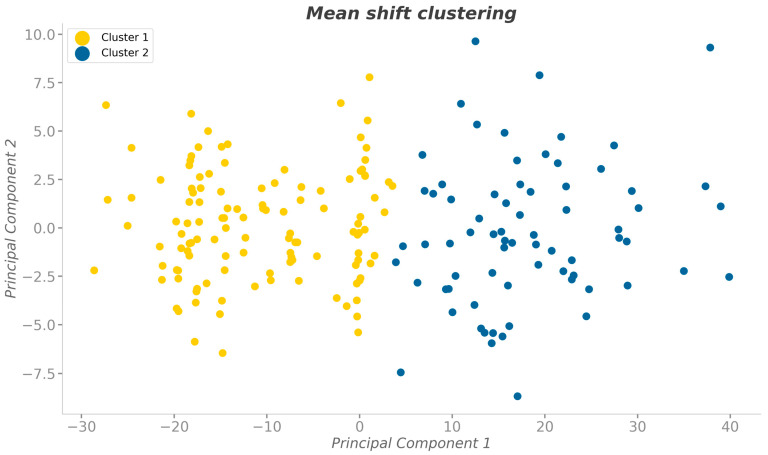
Representation of data distribution from the application of the clustering algorithms within the space of the first two principal components.

**Table 1 sensors-24-02929-t001:** Results of the postural analysis using ML algorithms with sex differences and test–retest reliability.

	Postural Parameters	Mean ± SD		Sig.	Effect Size (d)	ICC	SEM	MDC
		*Men*	*Women*					
Body joints (°)	Shoulder adduction angle	16.1 ± 1.9	14.1 ± 1.5	<0.001 ***	1.14	0.94	0.22	0.61
	Elbow extension angle	7.6 ± 3.6	4.4 ± 2.1	<0.001 ***	1.07	0.93	0.60	1.67
	Hip adduction angle	9.9 ± 2.2	6.7 ± 1.5	<0.001 ***	1.67	0.95	0.16	0.45
	Hip extension angle	3.4 ± 2.3	2.5 ± 1.7	0.005 **	0.50	0.78	0.81	2.25
	Knee varus/valgus angle	2.6 ± 1.0	2.2 ± 0.9	0.027 *	0.39	0.93	0.17	0.44
	Knee extension angle	2.7 ± 1.7	2.7 ± 1.8	0.906	0.01	0.84	0.68	1.89
	Ankle flexion angle	68.6 ± 5.3	72.9 ± 4.9	<0.001 ***	−0.85	0.85	0.67	1.85
Horizontal inclinations (°)	Ear line	2.0 ± 1.5	2.0 ± 1.2	0.550	0.02	0.79	0.49	1.38
	Shoulder line	1.2 ± 0.7	1.2 ± 0.9	0.740	−0.01	0.73	0.48	1.33
	Elbow line	1.2 ± 0.9	1.3 ± 0.9	0.530	−0.12	0.85	0.33	0.93
	Wrist line	1.3 ± 0.9	1.5 ± 0.9	0.408	−0.13	0.83	0.34	0.95
	Hip line	1.2 ± 0.8	1.5 ± 1.0	0.071	−0.34	0.84	0.36	1.01
	Knee line	2.2 ± 1.3	2.1 ± 1.4	0.692	0.04	0.67	0.83	2.30
	Ankle line	1.9 ± 1.4	2.0 ± 1.3	0.581	−0.08	0.80	0.67	1.87
Vertical inclinations (°)	Neck inclination	13.6 ± 3.2	15.4 ± 3.3	<0.001 ***	−0.55	0.93	0.89	2.47
	Trunk forward inclination	2.3 ± 1.4	1.5 ± 1.1	<0.001 ***	0.66	0.77	0.43	1.20
	Body imbalance	0.9 ± 0.4	1.3 ± 0.6	<0.001 ***	−0.64	0.90	0.12	0.35
	Leg inclination	1.8 ± 0.6	1.8 ± 0.6	0.547	−0.09	0.80	0.23	0.64
Vectors length (pd)	Shoulder–hip difference	83.8 ± 14.9	63.4 ± 13.3	<0.001 ***	1.44			
	Torso vector	292.3 ± 26.3	244.7 ± 29.5	<0.001 ***	1.71			
	Total arm vector	297.5 ± 33.4	257.3 ± 32.9	<0.001 ***	1.21			
	Total leg vector	388.7 ± 32.4	358.2 ± 32.4	<0.001 ***	0.94			

SD = Standard deviation, pd = pixel distance. Significance levels: * *p* < 0.05, ** *p* < 0.01, *** *p* < 0.001. Cohen’s d: >0.50 = medium effect size, >0.80 = large effect size. ICC = intraclass correlation coefficient (3,k). SEM = standard error of the mean. MDC = minimum detectable change.

**Table 2 sensors-24-02929-t002:** Results of the postural analysis with ML algorithms based on the mean-shift clustering.

	Postural Parameters	Mean ± SD		Sig.	Effect Size (d)
		*CG1*	*CG2*		
Body Joints (°)	Shoulder adduction angle	14.7 ± 1.6	15.1 ± 2.2	0.112	0.24
	Elbow extension angle	4.8 ± 2.6	6.3 ± 3.4	0.001 **	0.50
	Hip adduction angle	6.5 ± 1.6	9.4 ± 2.2	<0.001 ***	1.53
	Hip extension angle	2.5 ± 1.6	3.2 ± 2.4	0.035 *	0.36
	Knee varus/valgus angle	2.3 ± 0.9	2.5 ± 1.0	0.258	0.19
	Knee extension angle	73.1 ± 4.7	69.2 ± 5.5	0.146	−0.24
	Ankle flexion angle	14.7 ± 1.6	15.1 ± 2.2	<0.001 ***	−0.78
Horizontal inclinations (°)	Ear line	2.0 ± 1.2	2.0 ± 1.4	0.89	−0.02
	Shoulder line	1.3 ± 0.9	1.1 ± 0.7	0.032 *	−0.34
	Elbow line	1.3 ± 1.0	1.2 ± 0.8	0.591	−0.08
	Wrist line	1.4 ± 0.9	1.4 ± 0.9	0.958	−0.01
	Hip line	1.7 ± 1.0	1.1 ± 0.8	<0.001 ***	−0.64
	Knee line	2.2 ± 1.4	2.1 ± 1.4	0.418	−0.12
	Ankle line	2.0 ± 1.4	2.0 ± 1.4	0.822	0.04
Vertical inclinations (°)	Neck inclination	15.5 ± 3.1	14.0 ± 3.4	0.006 **	−0.46
	Trunk forward inclination	1.5 ± 1.1	2.0 ± 1.4	0.007 **	0.42
	Body imbalance	1.3 ± 0.6	1.0 ± 0.4	0.004 **	−0.51
	Leg inclination	1.9 ± 0.6	1.7 ± 0.6	0.034 *	−0.31
Vector length (pd)	Shoulder–hip difference	57.5 ± 7.0	86.6 ± 10.7	<0.001 ***	3.21
	Torso vector	233.4 ± 16.2	295.9 ± 21.6	<0.001 ***	3.28
	Total arm vector	241.2 ± 14.1	307.7 ± 23.4	<0.001 ***	3.44
	Total leg vector	342.6 ± 19.4	400.3 ± 22.2	<0.001 ***	2.77

CG1 = clustering group 1; CG2 = clustering group 2. SD = standard deviation, pd = pixel distance. Significance levels: * *p* < 0.05, ** *p* < 0.01, *** *p* < 0.001. Cohen’s d: >0.50 = medium effect size, >0.80 = large effect size.

## Data Availability

The datasets used and/or analyzed during the current study are available from the corresponding author on reasonable request.

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
