# Peer review of "Biomechanical Posture Analysis in Healthy Adults with Machine Learning: Applicability and Reliability"

_sensors, 2024, doi:10.3390/s24092929_

Round 1

Reviewer 1 Report

Comments and Suggestions for Authors

The authors examine the use of ML for posture assessment of young men and women, and found sex-specific postural differences, with measures demonstrating high reliability, and clustering analysis identifying shoulder-hip distance for distinguishing clusters. While findings are of interest a few questions remain.

Introduction.

1. While few studies are noted to have been focused on posture assessment, a discussion of how the current study compares and contrasts with recent work examining computer vision and applications of posture/gait assessment would be beneficial to readers. [1-4]

Materials and Methods.

2.  Did the authors consider normalization of data, given expected anthropometric differences between cohorts?

Results.

3. What were the heights, weights, and ages of men and women included in study? Any significant differences? If so, how would they impact postural parameters collected?

4. Were observed gender differences in postural parameters driven by changes in height? Please be sure to assess correlations between postural parameters with anthropometric measures, and discuss implications in discussion.

Discussion.

5. Be sure discuss how length of body vectors are quantitatively associated with body lengths, and not just qualitatively as currently mentioned in lines 258-259.

6. What are latent features that the clusters are associated with to help support statement that use of PCA and clustering suggest that posture classification may be possible regardless of sex? As a follow-up, exactly how would this method be applicable to applications noted on lines 284-286?

Literature Cited:

MassirisFernández, Manlio, et al. "Ergonomic risk assessment based on computer vision and machine learning." Computers & Industrial Engineering 149 (2020): 106816.
Zhang, Zhuoyu, et al. "Automated and accurate assessment for postural abnormalities in patients with Parkinson’s disease based on Kinect and machine learning." Journal of NeuroEngineering and Rehabilitation 18 (2021): 1-10.
Kaur, Rachneet, et al. "A Vision-Based Framework for Predicting Multiple Sclerosis and Parkinson's Disease Gait Dysfunctions—A Deep Learning Approach." IEEE Journal of Biomedical and Health Informatics 27.1 (2022): 190-201.
Zahra, Syeda Binish, et al. "Marker-based and marker-less motion capturing video data: Person and activity identification comparison based on machine learning approaches." Computers, Materials & Continua 66.2 (2021): 1269-1282.

Reviewer 2 Report

Comments and Suggestions for Authors

Round 2

Reviewer 1 Report

Comments and Suggestions for Authors

The authors have adequately addressed all questions.